# Characterization and Therapeutic Applications of Biosynthesized Silver Nanoparticles Using *Cassia auriculate* Flower Extract

**DOI:** 10.3390/plants12040707

**Published:** 2023-02-05

**Authors:** Nadana Sabapathi, Srinivasan Ramalingam, Kandasamy Nagarajan Aruljothi, Jintae Lee, Selvaraj Barathi

**Affiliations:** 1Guangdong Key Laboratory for Genome Stability and Disease Prevention, School of Medicine, Shenzhen University, Shenzhen 518060, China; 2Department of Horticulture & Life Science, Yeungnam University, Gyeongsan-si 38541, Republic of Korea; 3Department of Genetic Engineering, SRM Institute of Science and Technology, Kattankulathur, Chennai 603 203, India; 4School of Chemical Engineering, Yeungnam University, Gyeongsan-si 38541, Republic of Korea

**Keywords:** biosynthesis, *Cassia auriculate*, silver nanoparticles, antimicrobial, live/dead cells, antioxidant

## Abstract

The current study analyzes the biosynthesis of silver nanoparticles using the *Cassia auriculate* flower extract as the reducing and stabilizing agent. The *Cassia auriculate*- silver nanoparticles (Ca-AgNPs) obtained are characterized by UV–Vis spectroscopy, Fourier transform infrared (FT-IR) spectroscopy, X-ray diffraction (XRD), scanning electron microscopy (SEM), and transmission electron microscopy (TEM) analysis. The results of the spectral characterization have revealed that the surface Plasmon resonance band observed at 448 nm confirms the formation of AgNPs. TEM analysis of the Ca-AgNPs was a predominately spherical shape with a size assortment of 30 to 80 nm and an angular size of 50 nm. The well-analyzed Ca-AgNPs were used in various biological assays, including healthcare analysis of antimicrobial, antioxidant (DPPH), and cytotoxic investigations. Ca-AgNPs showed efficient free radical scavenging activity and showed excellent antimicrobial activity against to pathogenic strains. The occurrence of Ca-AgNPs lead to reduced Live/Dead ratio of bacteria (from 36.97 ± 1.35 to 9.43 ± 0.27) but improved the accumulation of bacterial clusters. The cytotoxicity of Ca-AgNPs was carried out by MTT assay against MCF-7 breast cancer cells and a moderate cytotoxic. The approach of flower extract-mediated synthesis is a cost-efficient, eco-friendly, and easy alternative to conventional methods of silver nanoparticle synthesis.

## 1. Introduction

Nanotechnology has emerged as a leading technology in several sectors with an ultimate application in the agriculture, food, and pharma industries and biomedicine engineering [1]. Nanoparticles, due to their small sizes, versatility, and ready to couple with optical, textile, magnetic, electronic, mechanical, and chemical substances made, the candidates for novel applications in therapeutics, for example, in antimicrobial, antioxidant, and cancer cytotoxic properties [2]. In addition, nanoparticles have been used extensively in physical, chemical, and biological sciences. Although physiochemical approaches are employed in synthesizing nanoparticles, they are often related to harmful effects. Therefore, seeking an alternate method to encounter these disadvantages is more critical. Recently, several successful reports affirmed the production of these nanomaterials from natural sources such as plants and microbes [3]. The biosynthesis of nanocomposites is a large-scale scientific domain with significant attention in biomedical applications due to their biocompatibility and multifunctional abilities [4]. Importantly, plant-based synthesis of metal nanoparticles is getting boundless consideration due to their availability, eco-friendly nature, less time consumption and low toxicity [5], and creation of essential nanoparticles with predefined highlights.

The viral illness that caused the current pandemic has increased human fear and is destroying the planet. Globally, people lost their lives, while others lost their families, their jobs, and their access to a quality education, all of which contributed to global economic crises. The primary cause of this epidemic is the coronavirus [6]. Other viruses, such as those that cause the life-threatening diseases, HIV, Herpes, Influenza, Hanta, Ebola, and Nipah, also grow and spread widely in addition to the coronavirus [7]. To create vaccines against the virus, pharmaceutical companies from all over the world are working with researchers. However, the world still struggles to accept it. This alarms for urgent antiviral drug development to protect human health from life-threatening viruses. 

Metallic nanoparticles are gaining popularity because of their distinctive properties and applications. Silver metal nanoparticles are the most extensively studied because they have tremendous broad-spectrum activities. Various biosynthesized nanoparticles, silver nanoparticles are desirable because of their biological, physical, and chemical properties [8]. Metallic nanoparticles used in several biomedical applications, silver nanoparticles are one of the most important and fascinating nanomaterials [9,10]. Notably, silver nanoparticles (AgNPs) can act as an antimicrobial and antiseptic agent and control DNA replication and cell division [11]. Ultimately, the AgNPs have been documented for their cytotoxicity in numerous cancer cell lines, with their ability to control the physiological roles of cancer cells and induce DNA damage in turn apoptosis in vitro [12]. Novel bio-nano-formulations make ineffective drugs better by modifying their delivery and uptake rates. Using plants as a source in biological nanoparticle manufacturing has a benefit because they include many mediator molecules such as reductase and dehydrogenases, which could be crucial in the generation of attractive nano-formulations [13]. Therefore, the current study focused on synthesizing silver nanoparticles using the extracts of *Cassia auriculate* flower and tested the medicinal properties of the nanoparticles. *Cassia auriculate* is a medicinal plant extensively used in ayurveda, readily found primarily in India from the *Fabaceae* family and the *Senna genus.* The investigation of *C. auriculate* flower extract in combination with silver nanoparticles is unique and very rarely reported before.

This study aims to highlight the biosynthesized silver nanoparticles and describing how they are characterized and their therapeutic application in the field of biomedical (Figure 1). Silver ions were reduced and silver nanoparticles were formed in the present experiment when Ca-AgNPs leaf extract was added to a strong aqueous solution of silver nitrate and the resulted biosynthesized nanoparticles were characterized by ultraviolet-visible spectroscopy (UV-Vis), Fourier transform infrared (FTIR), X-ray diffraction (XRD), Scanning Electron microscopy (SEM), and Transmission Electron microscopy (TEM) analysis. It was expected that the CA bark extract acts as a capping agent to enhance the basic biomedical analysis of antimicrobial, antioxidant, and cytotoxic activities potential of the Ca-AgNPs properties were evaluated.

## 2. Results and Discussion

### 2.1. UV-Vis Spectral Study of Ca-AgNPs

The most simple and indirect technique, UV-Vis spectroscopy, was used to assess the bioreduction of silver ions to Ag nanoparticles. The fresh flower extract can transform silver nitrate into silver nanoparticles when the section of *Cassia auriculate* flower is added. The color change from reddish brown to yellow indicates nanoparticle formation, which, in our case, was attained after 24 h of incubation (Figure 1). The presence of Ca-AgNPs was confirmed via UV-visible spectroscopy, where it showed the maximum absorbance band at about 448 nm, which is a clear indication of the formation of AgNPs in the corresponding solution due to the surface plasmon resonance (SPR) [14]. Recently, a study reported that biosynthesized silver nanoparticles produced the same results in various plant extracts, including *Nigella sativa* seed extracts and *Boerhavia erecta, Decaschistia crotonifolia*, and *Aerva lanata* flower extracts [4].

### 2.2. FTIR Analysis of Synthesized Ca-AgNPs

Fourier transforms infrared (FTIR) spectroscopy helps find the functional groups that participated in capping Ca-AgNPs and reduction. The FTIR band of the Cassia auriculate flower extract displayed leading absorption bands at 3455, 1672, 1428, 1109, 850, and 619 cm^−1^ (Figure 2A). The FTIR band of Ca-AgNPs showed the primary adsorption at 3392, 1527, 1361, 1032, 832, and 647 cm^−1^, which indicates the presence of phytoconstituents that can play as capping representatives. The FTIR spectra of both Cassia auriculate leaf extracts and biosynthesis of Ca-AgNPs displayed the main and slight changes of the bands rationally because of the capping and difference in the stability of the produced nanoparticles [15]. A change is detected for the bands at 3455 cm^−1^ to a slight wavelength of 3392 cm^−1^ due to the contribution of N-H bonding extended by phenolic complexes found in the leaf extract [16]. The following band at 2995 cm^−1^ is owing to the C−H extension of the aliphatic cluster or methylene cluster and a distinctive band of triterpenoid saponins [17].

The band at 1672 cm^−1^ shifted to a lower wavelength of 1527 cm^−1^ displaying the connection of aromatic C=C stretch; the peak at 1428 cm^−1^ in extract shifted to the lower wavelength at 1361 cm^−1^ in the Ca-AgNPs sample, indicating the occurrence of the −C−O extending of tertiary alcohols; the absorption peak at 1109 cm^−1^ shifted to the lower wavelength of 1032 cm^−1^ shows the O−H growing of the phenol cluster [18]; the band at 850 and 832 cm^−1^ was appeared as of the connection of aliphatic chloro- compounds or C−S stretch [19]; and the peaks at 619 cm^−1^ shifted into higher wavelength 647 cm^−1^ showing the aromatic group of C−H stretching. The results revealed that most of the peaks correspond to the several phytoconstituents compounds such as polyphenols, steroids, alkaloids, tannins, and triterpenoids, which sufficiently appeared in the leaf extract, which used to reduce the silver nitrate to Ca-AgNPs. These findings from the phytochemical analysis of the C. auriculate flower are inconsistent.

### 2.3. XRD Analysis of Synthesized Ca-AgNPs

Biosynthesized Ca-AgNPs were characterized further by X-ray diffraction methods, showing various apices at 2θ such as 31.102, 38.115, 42.193, 46.814, 57.275, and 65.102 (Figure 2B). These leading picks in the band, matching the 111, 200, 211, 120, 202, and 220 planes, individually, reflect the crystalline and face-centered cubic (fcc) patterns of the biosynthesized AgNPs. Both peaks indicated the crystal structure of biosynthesized silver nanoparticles. Additionally, the peak at (120) was equivalent to the AgNPs synthesized from Excoecaria agallocha. Several other researchers have also corroborated the same peak to silver nanoparticles [20,21]. The dimensions of the nanoparticles will essentially impact the XRD patterns [22], where the occurrence of different droppings in peaks represents the interference by the extract, which provides stability and crystal nature to the AgNPs, which was studied well in other biosynthesized nanoparticles [23].

### 2.4. Morphological Analysis of Synthesized Ca-AgNPs

The structural properties of the biosynthesized Ca-AgNPs were investigated by SEM analysis. The SEM results clearly revealed that the prepared Ca-AgNPs were smooth and spherical with relatively uniform size and appeared as loose clusters (Figure 3A). These show the AgNPs’ stability and inform the reader that AgNPs have a spherical morphology and nanoscale dimensions [24,25]. Similar results were reported in previous studies where the AgNPs were biosynthesized using Cucumis prophetarum extracts [1], with an average size of 54 ± 2.8 nm. The clustering of the Ca-AgNPs was probably encouraged by the dehydration step applied during the preparation of the samples for SEM analysis. We further assessed the size and shape of the Ca-AgNPs using TEM analysis, and the results confirmed that the size of the Ca-AgNPs ranged from 30 to 80 nm. A higher proportion was 50 nm, and most of the particles appeared spherical, and few were polygonal in shape (Figure 3B). Indeed, other researchers [26] and [27] have also reported similar results regarding the shape and size of the nanoparticles from the TEM analyses. The smaller size and spherical shape exhibited by the Ca-AgNPs are generally desirable for biomedical applications such as drug delivery. A specific area of the spreading design from the TEM analysis symmetrically displays a sharp area around the optimistic areas demonstrating the crystal-like nature of Ca-AgNPs.

### 2.5. Antibacterial Properties

The antibacterial activity of the *Cassia auriculate* flower extract and its biosynthesized Ca-AgNPs was determined through the disc diffusion process in one gram-positive strain, *S. epidermidis* and three gram-negative strains *P. aeruginosa, V. cholera* and *E. coli* strains. The antibacterial activity was assessed by measuring the zone of inhibition against these strains after 24 h of treatment. The results showed that all the strains tested exhibited higher sensitivity in Ca-AgNPs treatment compared to *Cassia auriculate* flower extract (Figure 4). The highest zone of inhibition observed at 75 µg/mL concentration of Ca-AgNPs was 16 and 13 mm in *V. cholerae* and *S. epidermidis* strains, respectively. Ca-AgNPs displayed better activity against *V. cholera* than all other bacterial strains (Table 1). The commending antibacterial activity of *Cassia auriculate* flower extract is not surprising, as other researchers have also demonstrated in their [28]. Generally, the silver nanoparticles exhibit remarkable antimicrobial activity in gram-negative bacteria, which is well documented in previous studies, for example, the reports by Jagtap et al. [29].

### 2.6. Antioxidant Properties

Reactive oxygen species are the main reason for oxidative stress associated with cancer, mental disease, aging, also neurodegenerative problems. Therefore, the researchers utilize the usage of bioactive compSounds as normal antioxidants to defend human suffering against oxidative stress. In this study, the antioxidant activity of Cassia auriculate flower extract and Ca-AuNP was assessed by DPPH assays; the ascorbic acid was used as a standard. The results indicated that the extract and Ca-AgNPs showed concentration-dependent scavenging activity to DPPH (Figure 5). For the Cassia auriculate flower extract and Ca-AgNP, the maximum scavenging activity is 71.8 ± 1.6% and 94.8 ± 2.18%, respectively, at 150 µg/mL concentration. The scavenging action of Ca-AgNPs is nearly equal to the standard ascorbic acid at 150 µg/mL concentration. The outcome revealed that Ca-AgNPs have outstanding antioxidant activity, corroborated by several other studies [30,31]. 

### 2.7. Live/Dead Cells Analysis

The bacterial cells viability was evaluated with the different concentrations (20, 40, 60, 80, 100, and 120 µg/mL) of Ca-AgNPs. Figure 6A showed in the control (a) slide mainly composed of active live cells (green) and the number of dead cells was increased when the cells are treated with Ca-AgNPs. With increasing Ca-AgNPs concentrations (b–f), the proportion of live cells decreased. In addition, Figure 6B Ca-AgNPs toxicity strongly inhibits bacterial growth and microbial metabolism at the high concentrations (above 60 µg/mL) of Ca-AgNPs by producing reactive oxygen species and DNA damage [32]. Additionally, a protective response-dependent increase in cell aggregations was evident from concentrations of 60 mg-N/L along with a pronounced live cell reduction.

### 2.8. Cytotoxic Analysis

The cytotoxicity was tested in MCF-7 cells using an MTT assay with varied concentrations of Ca-AgNPs (0, 20, 40, 60, 80, 100 μL/mL). The cells were treated with the nanoparticles for 24 h, and the MTT assay was performed. The assays were carried out in triplicates and repeated three times. The MTT assay result displayed a substantial reduction in cell viability with an increase in concentration of Ca-AgNPs (Figure 7). The IC_50_ values of Ca-AgNPs for MCF-7 were found to be 50.45 μL/mL. Cytotoxicity of the biosynthesized Ca-AgNPs can be observed from lower concentrations as well, but at a lower rate the cytotoxicity is negligible. 

This assay revealed the important antiproliferative activity of Ca-AgNPs. Similarly, silver nanoparticles synthesized from the Cucurbita pepo L fruit leaf extracts, when treated with MCF-7 cancer cells, showed promising outcome [33]. According to previous reports, nanoparticles alone can exhibit better activities against many human cancer cell lines [34], whereas, in our study the activity is due to the combined effects of AgNP and phytocompounds of Cassia auriculate attached to the nanoparticle surface. Thus, it is important to investigate the molecular mechanisms behind the anticancer properties of the synthesized Ca-AgNPs. 

## 3. Materials and Methods

### 3.1. Preparation of the Extract

Fresh flowers of *Cassia auriculate* (20 g) were collected, washed, and heated with 100 mL of Millipore H_2_O for 10 min. Then, the samples were centrifuged, and the supernatant was separated, passed through Whatman filter paper, and kept refrigerated for further studies [35].

### 3.2. Synthesis of Silver Nanoparticles

The flower extract of *Cassia auriculate* was mixed with 0.3M of silver nitrate and subjected to sonication with an ultrasonic wash (PCI Ultrasonic 1.5 L (H)) at room temperature (37 °C). The solution was then shaken in dark at 35 °C until the silver nanoparticles were formed, as indicated by the change in color from reddish-brown to yellow (usually, it takes 24 h). The silver nanoparticle was re-dissolved in H_2_O, centrifuged at 10,000 rpm for 20 min, and the supernatant was further evaluated [36].

### 3.3. Characterization of Biosynthesized Silver Nanoparticles

Based on the prior literature concerning Ca-AgNP biosynthesis [37], the maximum level of absorbance for the synthesis of AgNPs was determined in the assortment of 300–800 nm wavelength through a spectrophotometer (Agilent CARY 60). Many factors moderate the features of AgNPs, such as size, morphological shape, crystal structure, surface charge, and biological activity. 

#### 3.3.1. UV–Vis Spectroscopy

UV-Vis spectroscopy is a straightforward approach for the primary characterization of AgNPs through a double-beam Perkin–Elmer lambda to analyze their optical properties [38] with a measurement range of 200–800 nm. The free-electron sways and creates charges over the outer layer of nanoparticles under electromagnetic radiations due to the SPR impact [39]. *Cassia auriculate* flower extract biosynthesized nanoparticles have shown a broad-spectrum curve in UV-Vis analysis. Similarly, several studies have used the UV-Vis analysis and have established the changes in the reaction color and decreased Ag ions resulting in nanoparticle synthesis.

#### 3.3.2. Fourier Transform Infrared (FTIR)

FTIR is an excellent, insightful technique that recognizes chemical structure, functional groups, and bonding arrangements of molecules [40]. To identify the molecules that serve as coating and stabilizing agents and to spot the reduction of silver ions, AgNPs are characterized using FTIR [39]. The samples in this study were analyzed by a Vertex 80V FTIR system (Bruker, Germany) to obtain the infrared spectrum of Ca-AgNPs using the conduction method at a 2 cm^−1^ ranging from 4000–400 cm^−1^.

#### 3.3.3. X-ray Diffraction (XRD)

XRD was used to examine the crystalline structure and purity of the biosynthesized Ca-AgNPs. The atoms place themselves at an appropriate distance on the crystalline plane and show a form of diffraction [41]. The experiment was carried out on an X-ray diffractometer (DMAX-2500 XRD) (Rigaku, Tokyo, Japan) using a Nickel filter and Cu Kα (1.540 A°) radiation on 40 kV and 30 mA, and a perusing assortment used between 30 and 80°.

#### 3.3.4. Scanning Electron Microscopy (SEM)

SEM analysis provides a high-resolution image of the surface of nanoparticles which uncovers information such as size, composition, shape, electrical conductivity, topography, and other properties [37,42]. The morphology and size of Ca-AgNPs were evaluated by an SEM (Hitachi, S-4300SE, Japan), and the analyses were carried out using Vega TC software.

#### 3.3.5. Transmission Electron Microscopy (TEM) Analysis 

TEM analysis is an advanced technology that directly visualizes the image gained from the transmitted electrons. It provides high resolution of the chemical and structural facets of the nanoparticles [37]. The size and structure of the Ca-AgNPs were further resolved with TEM (JEM 2100F microscope, Jeol, Tokyo, Japan). The samples of Ca-AgNPs were arranged by placing a drop of AgNP solution on the copper-coated grid and aeration the pieces using a vacuum chamber [43]. The images of the Ca-AuNPs are shown with a scale bar of 100 nm. Inter planar spaces of crystal exposed by high-resolution TEM (HRTEM) were resolute by micrograph of digital analysis (3.0 Gatan Version). This method is one of the trustable techniques to analyze the formation of AgNPs by directly visualizing the image of the nanoparticles to find the shape, size, diameter, and core structure [41].

### 3.4. Antimicrobial Analysis of Ca-AgNPs

The antimicrobial activity of biosynthesized Ca-AgNPs was studied with four bacterial strains using a disc diffusion method. Gram-positive *Staphylococcus epidermidis* and three Gram-negative strains of *Pseudomonas aeruginosa*, *Vibrio cholera,* and *Escherichia coli* strains were cultured in the nutrient broth at 37 °C for 24 h to obtain the bacterial suspensions. The cultures were spread in nutrient agar plates to obtain a lawn of bacterial growth. Immediately after spread plating, 5 mm of purified Whatman filter disc with diverse concentrations of biosynthesized Ca-AgNPs was placed on the plate to obtain a zone of inhibition by Ca-AgNPs, with which the antimicrobial property was calculated.

### 3.5. Antioxidant Activity

To evaluate the antioxidant property of the Ca-AgNPs, the antioxidant assay using 1, 1-diphenyl-2-picryl-hydrazine (DPPH) was carried out as described by McDonald et al. [43]. The outcomes were demonstrated using various concentrations (25, 50, 75, 100, 125, and 150 µg/mL) of Ca-AgNPs and compared with extract only (negative control) and the ascorbic acid (positive control). The radical scavenging activity by butylated hydroxytoluene (BHT) was used as a standard.

### 3.6. Live/Dead Cells Staining Assess

The live and dead cells in the biofilm communities were stained with the fluorescent n dyes SYTO9 and PI from a Live/Dead BacLight bacterial viability kit (7012, Molecular Probes, Leiden, the Netherlands). Generally, the bacterial culture suspension (125 L) was added to tubes containing 50 L of SYTO9 and PI assorted solution, incubated at 20 °C in the dark for 15 min, and then qualitative images were captured using a fluorescence microscope (Nikon Eclipse E600, Tokyo, Japan). After that, ImageJ (NIH, Madison, WI, USA) was used to quantify both the live and dead bacteria from the fluorescent images by randomly choosing different spots (n = 3) on the films. The percentage of green to total fluorescence (red + green) was thought to be equivalent to the proportion of living cells to all other cells in the biofilm (alive + dead). Additionally, the changes in EPS composition and characteristics were used to evaluate the effects of nitrite on the bacterial cells. 

### 3.7. Assessment of Cells Viability Study by MTT Assay

The MTT assay was used to assess the cytotoxic potential of Ca-AgNPs in breast cancer cell lines. The cancer cells were cultured on flasks with Dulbecco’s Modified Eagle’s Medium (DMEM) containing 10% fetal bovine serum (FBS) and 1% penicillin and streptomycin in a CO_2_ incubator for 24 h at 37 °C. Following the incubation periods, cell lines were tested with various concentrations of biosynthesized Ca-AgNP (0, 20, 40, 60, 80, and 100 µg/mL), and untreated cells were used as a control in a 96-well plate. Post-treatment, the cells were incubated in a CO_2_ growth chamber for 24 h. Under a microscope, the change in structure and morphology of the cells were observed. A 20 μL of MTT reagent (5 mg/mL) was added to each well, and the plate was incubated for 4 h at 37 °C, followed by 100 μL of DMSO to dissolve the formazan crystals that had formed. The percentage of cell viability was assessed by measuring the plate absorbance in an ELISA reader at 570 nm, which was utilized as the reference wavelength for using a 96-well plate. The assay was conducted in triplicates and repeated three times. We have expressed the data about the viability of the cells, which is obtained by using the following formula
*Cell viability (%) = 100 − [(Avg absorbance of control − Avg absorbance of treated) × 100].*

## 4. Conclusions

The current study determined that *Cassia auriculate* flower extract used as a stabilizing agent in the combination of an active and constant AgNPs, which have incredible usage in basic biomedical research. SEM and TEM analysis showed that Ca-AgNPs were spherical shaped with an average size around 50 nm. The biosynthesized Ca-AgNPs exposed admirable antimicrobial efficiency against to Gram-positive *S. epidermidis* and three Gram-negative strains *P. aeruginosa*, *V. cholerae,* and *E. coli* bacteria, and competent free radical scavenging activity. Ca-AgNPs showed efficient cytotoxic activity against to MCF-7 breast cancer cells and it could be further developed into a feasible anticancer medicine for cancer treatment. Together, the results suggest that Ca-AgNPs can be effectively used as anticancer agents, antimicrobial, antioxidants in medicine and food industries.

## Data Availability

Not applicable.

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
