# Peer review of "Characterization and Therapeutic Applications of Biosynthesized Silver Nanoparticles Using Cassia auriculate Flower Extract"

_plants, 2023, doi:10.3390/plants12040707_

Round 1
Reviewer 1 Report
The manuscript of “Characterization and therapeutic applications of biosynthesized
silver nanoparticles using Cassia auriculate flower extract” is focus on the Biosynthesis of silver nanoparticles using the Cassia auriculate flower extract as the reducing and stabilizing agent. This article highlights the biosynthesized silver nanoparticles and its therapeutic application in the field of biomedical field. This biosynthesized Ca-AgNPs showed antimicrobial efficiency against to Gram-positive S. epidermidis and three Gram-negative strains P. aeruginosa, V. cholerae and E.coli bacteria, and competent free radical scavenging activity. Also, its showed efficient cytotoxic activity against MCF-7 breast cancer cells. Author suggesting the approach of flower extract-mediated synthesis is a cost-efficient, eco-friendly, and easy alternative to conventional methods of silver nanoparticle synthesis. This manuscript is interesting, however, this manuscript needs revision. The following comments need to be addressed
Comments
· There are many similar articles available. Ex. “Green synthesis, characterization and catalytic activity of silver nanoparticles using Cassia auriculata flower extract separated fraction”” An investigation on cytotoxic effect of bioactive AgNPs synthesized using Cassia fistula flower extract on breast cancer cell MCF-7”” Biogenic synthesis of silver nanoparticles from Cassia fistula (Linn.): In vitro assessment of their antioxidant, antimicrobial and cytotoxic activities” etc.. author need to justify significance and novelty of the article
· Why “Cassia auriculate” is chosen for this study,
· How the author chooses this combination? And Why?
· How are the flowers processed, author used only the petals or whole flowers ?
· Section 3.1 : How much supernatant was obtained and how it is stored (Storage temperature)
· Section 3.1-3.3: authors have not cited any reference !
· Line no 348: what kind of basic biomedical research.? Explain in the introduction or discussion.
· Author mentioned “ it can be efficiently used in “food industries”, How? Discuss in detail with reference.
· There are some typos and grammatical errors. The article needs to be revised with the scientific editor.
· References should be cited as per MDPI citation format.
· Enhance the quality of images and update the recent references
Author Response
Detailed responses to the Reviewers’ comments
Thank you very much for your kind comments. We will try our best to revise the manuscript in order to improve its quality according to your suggestions and suit the requirements of the journal. Below are the itemized responses from authors to the comments of editor/reviewers.
(BLUE – Comments; Black – Authors response; Red – Revised text).
Reviewer #1:
The manuscript of “Characterization and therapeutic applications of biosynthesized silver nanoparticles using Cassia auriculate flower extract” is focus on the Biosynthesis of silver nanoparticles using the Cassia auriculate flower extract as the reducing and stabilizing agent. This article highlights the biosynthesized silver nanoparticles and its therapeutic application in the field of biomedical field. This biosynthesized Ca-AgNPs showed antimicrobial efficiency against to Gram-positive S. epidermidis and three Gram-negative strains P. aeruginosa, V. cholerae and E.coli bacteria, and competent free radical scavenging activity. Also, its showed efficient cytotoxic activity against MCF-7 breast cancer cells. Author suggesting the approach of flower extract-mediated synthesis is a cost-efficient, eco-friendly, and easy alternative to conventional methods of silver nanoparticle synthesis. This manuscript is interesting; however, this manuscript needs revision. The following comments need to be addressed
Comments
There are many similar articles available. Ex. “Green synthesis, characterization and catalytic activity of silver nanoparticles using Cassia auriculata flower extract separated fraction”” An investigation on cytotoxic effect of bioactive AgNPs synthesized using Cassia fistula flower extract on breast cancer cell MCF-7”” Biogenic synthesis of silver nanoparticles from Cassia fistula (Linn.): In vitro assessment of their antioxidant, antimicrobial and cytotoxic activities” etc.. author need to justify significance and novelty of the article.
Thanks for your valuable comments, the main research of this paper is spherical and polydisperse AgNPs as reducing, stabilizing, and capping agents were synthesized using the biological based of Cassia fistula flower extract (CA) with silver nanoparticles bark. Additionally, various methods were used to assess the physicochemical characteristics of the biosynthesised AgNPs.
- Why “Cassia auriculate” is chosen for this study,
We have chosen this plant of Cassia auriculate flower, due to the medicinal values of the plants used in many traditional medicinal systems including Ayurveda and Chinese traditional medicine and people use Cassia auriculata for diabetes, pink eye, joint and muscle pain (rheumatism), constipation, and other conditions and commercial availability.
- How the author chooses this combination? And Why?
The large levels of medicinal applications, biological activity and very less scientific evidences are available in this combination. Based on these reasons, we have selected combination.
- How are the flowers processed, author used only the petals or whole flowers?
For further analysis, the Cassia auriculate, flowers were collected, cleaned, cut into small pieces, dried at room temperature (28°C), and then ground into a fine powder. Using a rotavapor apparatus, the shade-dried leaf powder of Cassia auriculata was extracted with 70% ethanol under reflux for 8 hours and concentrated to a semisolid mass under reduced pressure. The yield was about 24% (w/w), and the material produced was a dark semisolid that was greenish-black in color. The study used the leftover extract, which was suspended in distilled water as needed. The air dried, powdered plant material was extracted in a conical flask at room temperature using separate amounts of petroleum ether, chloroform, alcohol, acetone-water, and water.
- Section 3.1: How much supernatant was obtained and how it is stored (Storage temperature)
The CA bark extract had an insipid yellow color that eventually turned dark brown, indicating the creation of biogenic AgNPs. After the reaction, the AgNPs dark solution was centrifuged for 30 minutes at 10,000 rpm to separate the pellet from the supernatant. Supernatants were centrifuged at 12,000 rpm for 30 minutes to collect AgNPs after the pellets had been removed. AgNPs were maintained in freeze-dried CA bark extract powder until their physiochemical analysis and biological activities were performed. Until it was used, it was kept at 4°C
- Section 3.1-3.3: authors have not cited any reference !
We have added references as per the reviewer suggestions
- Line no 348: what kind of basic biomedical research.? Explain in the introduction or discussion.
It was expected that the CA bark extract acts as a capping agent to enhance the basic biomedical analysis of antimicrobial, antioxidant, and cytotoxic activities potential of the Ca-AgNPs properties were evaluated (84-86).
- Author mentioned “ it can be efficiently used in “food industries”, How? Discuss in detail with reference.
When you consider the impact that food-borne illnesses have on the public's health, microbial contamination and food spoilage are major issues in the food industry [1]. It is necessary to create active food packaging with antimicrobial properties using active biocidal substances in order to reduce the severity of these issues. This packaging may help to reduce food spoilage and microbial contamination as well as improve product quality. Food industries used organic acids, enzymes, and polymers as packaging materials (biodegradable and non-degradable). Metal or metallic oxide nanoparticles have recently been developed with advantages over organic and inorganic acids due to their resistance to the harshest processing conditions, such as exposure to high temperatures [2]. Therefore, the use of nanoparticles in the food packaging sector may provide potential solutions for the problem posed by products with limited shelf lives, enhancing their quality and preventing microbial adhesion. Due to their antimicrobial properties, metallic nanoparticles such as silver, magnesium oxide, copper oxide, zinc oxide, cadmium selenite/tellurite, titanium, and gold have been used in the food industry.
- Carbone, M.; Donia, D.T.; Sabbatella, G.; Antiochia, R. Silver nanoparticles in polymeric matrices for fresh food packaging. J. King Saud Univ. Sci.2016, 28, 273–279.
- Emamifar, A.; Kadivar, M.; Shahedi, M.; Solimanian-Zad, S. Effect of nanocomposite packaging containing Ag and ZnO on reducing pasteurization temperature of orange juice. J. Food Process. Preserv.2012, 34, 104–112
- There are some typos and grammatical errors. The article needs to be revised with the scientific editor.
Thanks for your valuable comments, all the above-mentioned comments are revised
- References should be cited as per MDPI citation format.
All the references are checked once again and corrected based on the MDPI format
- Enhance the quality of images and update the recent references
Thanks for your valuable comments, all the above-mentioned comments are revised
Reviewer 2 Report
In this paper the Authors analyzes the Biosynthesis of silver nanoparticles using the Cassia auric- 11 ulate flower extract as the reducing and stabilizing agent. The paper is interesting but not original.
Moreover, Authors do not evaluate the effect of nanoparticles aggregation which affect cell response.
I suggest to analyse nanoparticles aggretates and report their size distribution for each of the tested concentrations in MTT.
Author Response
Detailed responses to the Reviewers’ comments
Thank you very much for your kind comments. We will try our best to revise the manuscript in order to improve its quality according to your suggestions and suit the requirements of the journal. Below are the itemized responses from authors to the comments of editor/reviewers.
(BLUE – Comments; Black – Authors response; Red – Revised text).
Reviewer #2:
In this paper the Authors analyzes the Biosynthesis of silver nanoparticles using the Cassia auric- 11 ulate flower extract as the reducing and stabilizing agent. The paper is interesting but not original. Moreover, Authors do not evaluate the effect of nanoparticles aggregation which affect cell response. I suggest to analyse nanoparticles aggretates and report their size distribution for each of the tested concentrations in MTT.
In this work, we didn't show the flower extract as a reducing agent or stabilizing agent. We agree with the reviewer that we didn't observe the nanoparticle aggregation in the cells, instead, we observed the effect of nanoparticle plus flower extract formulation on the cancer cells and reported the preliminary results that this formulation had the cytotoxicity. We also discussed that this phenomenon needs further investigation, which will be carried out in our future studies.
Round 2
Reviewer 2 Report
The paper can be accepted for publication.